# Polymorphisms within the *Boule* Gene Detected by Tetra-Primer Amplification Refractory Mutation System PCR (T-ARMS-PCR) Are Significantly Associated with Goat Litter Size

**DOI:** 10.3390/ani9110910

**Published:** 2019-11-01

**Authors:** Xiaoyue Song, Jie Li, Panfeng Fei, Xiaoyan Zhang, Chuanying Pan, Hong Chen, Lei Qu, Xianyong Lan

**Affiliations:** 1Shaanxi Provincial Engineering and Technology Research Center of Cashmere Goats, College of Life Science, Yulin University, Yulin 719000, China; songxiaoyue@yulinu.edu.cn; 2Key Laboratory of Animal Genetics, Breeding and Reproduction of Shaanxi Province, College of Animal Science and Technology, Northwest A&F University, Yangling 712100, China; lijie950302@163.com (J.L.); zhangxiaoyan1106@yeah.net (X.Z.); chuanyingpan@126.com (C.P.); chenhong1212@263.net (H.C.); 3College of Information Engineering, Northwest A&F University, Yangling 712100, China; f9187@163.com

**Keywords:** goat, *Boule* gene, litter size, association, tetra-primer amplification refractory mutation system PCR (T-ARMS-PCR), g.7254T>C

## Abstract

**Simple Summary:**

Mutations of the *Boule* gene, a gene that contributes to spermatogenesis, are a main cause of mammal infertility in reproduction. As a conserved gene, only some single nucleotide polymorphisms (SNPs) in the introns within it have been reported in humans with meiosis block, but not in goats, and especially its fundamental roles in female reproduction are still unknown. Therefore, this study detected the potential polymorphisms of the *Boule* gene in goats via the tetra-primer amplification refractory mutation system PCR (T-ARMS-PCR). Herein, one predicted SNP locus (rs661484476: g.7254T>C) of the *Boule* gene was first detected and displayed moderate polymorphism among all the studied groups. Notably, the polymorphisms of the goat *Boule* gene were significantly associated with the goat litter size in different groups, and female goats with the heterozygous genotype (CT) had more possibilities to produce multiple lambs than others, indicating that the *Boule* gene might underlie female reproductive capacities and that g.7254T>C could be a potential marker in the marker-assisted selection process for litter size in goats.

**Abstract:**

As a gene contributing to spermatogenesis, the *Boule* gene (also called *Boll*), whose mutations result in azoospermia and sterility of flies and mice, was conserved in reductional maturation divisions. However, in goats, the polymorphisms of *Boule*, especially with regard to their fundamental roles in female reproduction traits, are still unknown. Therefore, the aims of this study were to detect a potential mutation (rs661484476: g.7254T>C) located in intron 2 of the *Boule* gene by tetra-primer amplification refractory mutation system PCR (T-ARMS-PCR) and to explore its potential association with the litter size of Shaanbei White-Cashmere goats (SBWGs). In this study, g.7254T>C was firstly detected. The TT genotype was the dominant genotype in the single-lamb group, and T was also the dominant allele in all tested groups. Additionally, the detected locus displayed moderate polymorphism with polymorphism information content (PIC) values among all studied goats ranging from 0.303 to 0.344. Notably, according to the χ^2^ test, the distribution differences for the genotypic frequencies between the single- and multi-lamb groups was significant (*p* = 0.014). Furthermore, the polymorphisms of the goat *Boule* gene were significantly associated with the goat litter size in SBWGs (*p* < 0.05), which indicated that g.7254T>C could be a potential marker in the marker-assisted selection process for potential litter size in goats. These results also indicated that the *Boule* gene might exercise important functions in female goat reproduction, which provided new insight for female goat breeding and could accelerate the process of goat breeding.

## 1. Introduction

With high economic value, litter size traits will always be the most vital concern of the global goat industry. To meet increasing profitability demands and considering the low heritability of the goat reproductive trait, marker-assisted selection (MAS), which is based on the genetic diversity in reproductive potential, can accelerate the breeding process and improve reproductive efficiency [1,2]. A few genetic markers have been identified and associated with goat litter size, and our group has focused on them. Our previous study demonstrated that a novel 14 bp duplicated deletion within goat *GHR* and a 14 bp functional deletion within the *CMTM2* gene are significantly associated with litter size [3,4].

Similarly, the insertion/deletions (indels) in *SPEF2* [1] and the 16 bp indel of the lysine demethylase 6A gene (*KDM6A*) [5] were identified as significantly related to the first-born litter size in the Shaanbei White-Cashmere goat (SBWG) population. Additionally, the novel 26 bp indel within the catenin beta 1 gene (*CTNNB1*) has a significant relationship with goat litter size [6].

Furthermore, except for indel mutation, the main type of genetic variation, some single nucleotide polymorphisms (SNP) are proposed to adversely affect mRNA stability or protein binding and eventually change the target traits [7,8]. Based on whole genome scanning for the litter size trait in dairy goats (*Capra hircus*), various associated genes and SNPs were specifically selected, such as *SMAD2* and the aforementioned *KDM6A* [9]. Not only is it a candidate gene in goat reproduction, *SMAD2* has also been validated as being strongly associated with sheep prolificacy [10]. Another highly used multiparous gene, goat *GDF9*, was significantly associated with the first-born litter size in goats, and the famous Q320P mutation was the major SNP affecting goat litter size rather than V397I [11,12]. Additionally, c.682G>T and c.837T>C loci and diplotypes of the caprine *POU1F1* gene had significant effects on litter size (*p* < 0.05) [13]. Moreover, the missense mutation (L280V) within the *POU1F1* gene was also identified as strongly affecting growth traits and litter size in goats [14]. Furthermore, SNPs existing around genome flanking regions surrounding the transcription start sites also contributed to enhancing goat litter size [15].

As a complicated reproductive trait, litter size is controlled by multiple genes and factors [16], such as oocyte maturation, fertilization, embryogenesis, and spermatogenesis. Meanwhile, female sterility or male infertility, to a large extent, results in restricted dairy goat breeding [17]. As a confirmed member of the encoding germ-cell-specific RNA binding proteins, the *Boule* gene has been identified as being involved in regulating the development and differentiation of germ cells, playing essential roles in arrhenotoky [18,19,20]. Nevertheless, several studies have also explored the role of the *Boule* gene in female reproduction. For instance, in flatworm *Macrostomum lignano*, the *Boule* orthologues’ RNAi resulted in aberrant egg maturation and led to female sterility [21]. Moreover, the RNA and protein of *Boule* exhibit mitotic and meiotic expression in female Chinese sturgeon [22]. The process of embryonic stem cell differentiation to form germ-cell-like cells in a cumulus cell-conditioned medium induced the highest expression of the buffalo *Boule* gene [23]. However, in female goats, whether the *Boule* gene plays an important role in female reproduction is unknown.

As a conserved gene, *Boule* is considered to be of very low frequency in genetic variation. Only some SNPs in introns are reported in humans with meiosis block, but not in goats, especially in regard to their fundamental roles in female reproduction traits. Additionally, among various methods to detect SNPs, tetra-primer amplification refractory mutation system PCR (T-ARMS-PCR) has been successfully used in many studies because of its high efficiency [24]. Therefore, the objectives of this study were to detect the potential SNP of the goat *Boule* gene via T-ARMS-PCR in Shaanbei White-Cashmere goats and to investigate its correlation with goat litter size traits, which could accelerate the progress of goat breeding via an marker-assisted selection (MAS) strategy.

## 2. Materials and Methods

All experimental processing in this study was approved by the International Animal Care and Use Committee (IACUC) of the Northwest A&F University (protocol number NWAFAC1008). Additionally, the treatment of the experimental animals was fully consistent with local animal welfare guidelines, laws, and policies.

### 2.1. Animal Samples and Genomic DNA Collection

A total of 357 ear samples of Shaanbei White-Cashmere goats (SBWGs, female) were randomly collected from the Shaanbei Cashmere goat breeding farm in Shaanxi Province. All the used individuals were in the same age group (2–3 years old) and healthy. Moreover, from the agricultural technical station of Hengshan county by production records, data of the first birth litter size of Shaanbei Cashmere goats were also collected. According to the production records, the 357 goats were divided into a single-lamb group (*n* = 176) and a multi-lamb group (*n* = 181).

DNA samples were isolated from the ear tissues (saved in 70% alcohol at −80 °C) through the method of high salt extraction [25]. The purity of the DNA samples was assayed by a NanoDrop 1000 (Thermo Scientific, Waltham, Massachusetts, MA, USA). Then, all the DNA samples were diluted to 10 ng/L and kept at 4 °C temporarily.

### 2.2. Primer Design, Genotyped for g.7254T>C by T-ARMS-PCR

Referencing the goat *Boule* gene sequence (GenBank No: NC 030809.1; GeneID: 102173977), primers were designed to detect SNPs in the coding regions and noncoding regions of the *Boule* gene (Table 1) and were synthesized by Sangon Biotech (Shanghai, China).

Notably, the method of tetra-primer amplification refractory mutation system PCR (T-ARMS-PCR) was used in this study. With the use of specific inner and outer primers at exact proportion and amplification conditions, different alleles of the locus of interest were amplified. The specific inner and outer primers are shown in Table 1, and the specific steps are described below.

In the preliminary experiment, 30 SBWG individuals were randomly selected to construct a DNA pool for PCR amplification (pretest). Additionally, different groups were set up to explore the most suitable ratio of inner and outer primers and finally determine the most suitable reaction system (Table 2).

The pretest showed that the ratio of the inner and outer primers of the g.7254T>C was 1:4, which is the optimal ratio of inner and outer primers. Then the touchdown-PCR (TD-PCR) was performed as reported: initial denaturation for 5 min at 95 °C, followed by 18 cycles of denaturation for 30 s at 94 °C, annealing for 30 s at 68 °C (with a decrease of 1 °C per cycle), extension for 1000 bp/min at 72 °C, another 24 cycles of 30 s at 94 °C, 30 s at 50 °C, and 45 s at 72 °C, and a final extension of 10 min at 72 °C [26].

After that, the amplification products were detected by using 3.5% agarose gel electrophoresis. Different genotypes were distinguished according to the number and size of the fragments, and the two bands were homozygous, while the three bands were heterozygous. Furthermore, to confirm the mutation sites, DNA sequencing was performed only when products with different genotypes were first amplified. Subsequent genotyping of individuals was based on the results of the T-ARMS-PCR.

### 2.3. Statistical Analysis

After genotyping, analyses of the genetic polymorphisms, including gene frequencies, genotype frequencies, and polymorphism information content (PIC) were performed.

To test whether the polymorphisms were in Hardy–Weinberg equilibrium (HWE), the genotypic and allelic frequencies of the goat *Boule* gene were calculated directly using a chi-square (χ^2^) test [24]. Polymorphism information content (PIC) was calculated on the basis of Nei’s method, performed in the GDIcall Online Calculator (http://www.msrcall.com/Gdicall.aspx) [27]. Wherein, PIC > 0.5 means that the mutation is highly polymorphic, 0.25 < PIC ≤ 0.5 means moderate polymorphism, and PIC ≤ 0.25 means low polymorphism.

Distribution differences for genotypic and allelic frequencies between different groups were carried out with the χ^2^ test using SPSS software (Version 18.0) (IBM Corp., Armonk, NY, USA) [3].

Furthermore, the associations of the novel SNP of the *Boule* gene with goat litter size were inspected using the analysis of variance (ANOVA) available in SPSS (Version 18.0) [25], while different genotypes were considered as independent variables, and litter size traits were used as the dependent variable. The results were deemed to be statistically significant when *p* < 0.05, and all statistical tests were two-sided.

## 3. Results

### 3.1. Polymorphism Detection and Genotyping of Boule Gene via T-ARMS-PCR 

One SNP locus (NC-030809.1, rs661484476: g.7254T>C), located at the second intron of goat *Boule* gene, was firstly detected in goat groups (Figure 1 and Figure 2). The agarose gel electrophoresis after genotyping showed that the CC genotype was 364 bp and 521 bp, the CT genotype was 208 bp, 364 bp, and 521 bp, and the TT genotype showed 208 bp and 521 bp (Figure 1).

### 3.2. Analysis of Population Genetics of g.7254T>C

The numbers of genotypes, genotype frequency, allele frequency, PIC value, and HWE of the g.7254T>C were calculated (Table 3). The results showed that the TT genotype was the dominant genotype in the single-lamb group and that T was also the dominant allele. In the single-lamb group, the frequencies of wild genotypes (TT) were higher than the CT/ homozygous mutant genotype, while the heterozygous genotype (CT) had a higher frequency than the others in the multi-lamb group. Additionally, the results of the population parameters demonstrated that this SNP marker displayed moderate polymorphism, with a PIC among all studied groups ranging from 0.303 to 0.344.

Nevertheless, as for the Hardy–Weinberg equilibrium (HWE), the g.7254T>C locus in the multi-lamb group was not at HWE (*p* < 0.05), while the single-lamb group had the opposite result.

### 3.3. Association of g.7254T>C with Goat Litter Size

After the χ^2^ test (Table 4), the value of Pearson chi-square for the genotypic frequencies between different groups was 8.510 (df = 2), and the distribution difference was significant (*p* = 0.014), while the allelic frequencies between the single- and multi-lamb groups had no statistical difference (χ^2^ = 2.892, df = 1, *p* = 0.089).

To explore whether the polymorphisms of the goat *Boule* gene were related to the goat litter size, the associations between the g.7254T>C and litter size traits were investigated (Table 5). Consistent with the results of Table 4, individuals with the heterozygous genotype (CT) had higher frequencies of multi-lambs than goats with the TT genotype, indicating that the g.7254T>C had significant effects on litter size in the SBWG breed (*p* < 0.05) (Table 5). These results are also consistent with the distribution differences of the genotypic frequencies.

## 4. Discussion

In recent years, to better meet the growing market demand, molecular genetic breeding techniques, especially marker-assisted selection (MAS) based on genetic diversity, have been widely used in livestock breeding for their convenience and high efficiency [28]. As the most representative of genomic variation, the potential SNPs of candidate genes and the most effective method of detecting mutations will always be the most vital concern of MAS [29]. Among various detection methods, the T-ARMS-PCR is inexpensive, convenient, rapid, and accurate for SNP genotyping [24,30]. Therefore, in this study, the novel mutations of the *Boule* gene, an essential contributor of spermatocyte meiosis in reproduction, were genotyped by T-ARMS-PCR in goats.

Herein, mutations in the coding region of the *Boule* gene were not found, but one SNP locus (g.7254T>C) was detected at intron 2. The mutation rate of the *Boule* gene is lower than that of other genes, which may be caused by the lack of polymorphism in its exon sequence. Three nucleotide mutations in the *Boule* gene intron regions were previously detected in 164 infertile males but were not found in the coding regions and the 3′ untranslated regions [31]. Luetjens et al. [32] detected mutations in the 2~11 exons of the *Boule* gene in 18 patients with meiosis block, but no mutations were found in detected exons, which is consistent with the results of Westerveld et al. [33]. Low variation of the human *Boule* gene is the most likely constraint of its strong function, suggesting that the *Boule* gene is conserved to a great degree for meiosis.

In recent years, the function of introns has been continuously revealed, especially in the intron-mediated splicing process [34,35,36]. SNP is the main cause of abnormal splicing of exon skipping and intron retention [8]. It has been reported that mutations at the intron will activate the recessive variable splice sites [37]. On the other hand, introns contain many transcriptional regulatory elements, such as the intron splicing enhancer (ISE) and the intron splicing silencer (ISS) [37]. The *Boule* gene was found to be expressed in a similar pattern with gene of *DNA meiotic recombinase 1* and *mutS homolog 4* between 49 and 94 days postcoitum in the sheep ovary, suggesting its possible participation in female reproductive capacities [38].

The *Boule* gene is largely known as a main cause of sperm deficiency, but very little data exists concerning its potential function in female reproductive traits. In this study, g.7254T>C had a significant effect on litter size in the SBWG breed, speculating that the *Boule* gene might underlie female reproductive capacities. Concerning female gametogenesis in mammals, does the *Boule* gene exist purely to promote germ-cell progression? A large body of study of the *Boule* gene in the male meiosis process has been identified, but its potential function in females is largely unknown and awaits further deep study.

## 5. Conclusions

Taken together, the g.7254T>C of the goat Boule gene was found and shown to have a significant effect in the SBWG breed litter size. This means that g.7254T>C is an informative molecular marker in the MAS process for optimal litter size in goats and provides insights into new concepts in female reproduction.

## Figures and Tables

**Figure 1 animals-09-00910-f001:**
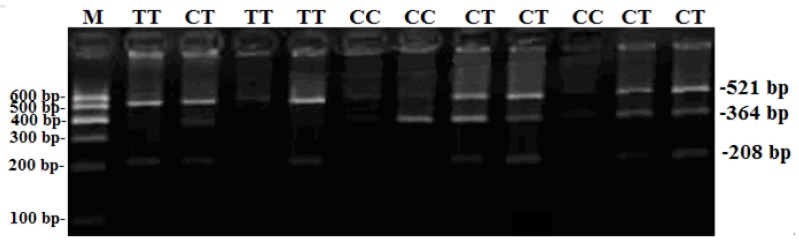
The sequencing map for g.7254T>C locus within the *Boule* gene.

**Figure 2 animals-09-00910-f002:**
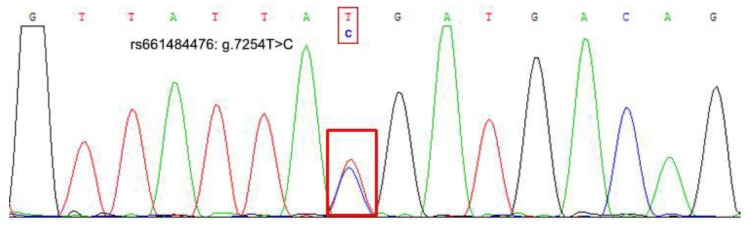
The sequencing map of the *Boule* gene’s g.7254T>C locus.

**Table 1 animals-09-00910-t001:** Single nucleotide polymorphism (SNP) primer information of the goat *Boule* gene.

Loci	Primer Sequences (5′→3′)	Tm (°C)	Sizes (bp)
P1	F: CGTGGTGTTGCTTCCTGGGTGT R: GGGAATCCTCAACGGCACAGAC	61.6	812
P2	F: TCAAATCAGACGCAAACAGA R: TTGGGATAAAGAAACAGGAC	47.8	568
P3	F: TAGAGTGACTGGTGGAAGCC R: CTGGTGGTCCAATGGTAAAG	50.1	884
P4	F: GTGCCTCAAAGAGTCGGAAAC R: CTGGGGTGGAGCTGATGTAATA	50.0	944
P5	F: AGCCAGCACTTCAACACTACAC R: CATTTGCCTACCACCTTCG	49.9	553
P6	F: GGACACGACTGAGCGACTGA R: GACCACCTGATGGGAAGAGC	51.3	797
P7	F: CAGTTCAGTCGCTCAGTCGT R: CTTACCCTCAACCTCCCATA	50.2	1265
P8	F: GTATCGGACTCTTCGCAACC R: GAAATGGTGAAGGACAGGGA	50.5	1191
P9	F: GGGGTCGCATAGAGTTGGA R: GGGGATTGAGTCAGGGATAGTT	50.3	636
P10	F: TCCTTCCAGCCATACCAAAC R: CCAGGGATTAAACTCAGACC	49.2	826
P11	F: AGTTTTCCGAGCACCACTTG R: CAGCTTCTAGCCGGTTCATT	50.6	935
P12	F: CACTGCCATGACTGGAGGA R: CTGCCAATTCAGGGGACA	50.5	818
P13	F: GGAGAAGGGAATGGCTAC R: CCTGATGATCTGAGGTGGA	48.2	978
P14	F: GTGAGTGCTCGCTGATAGT R: GGTGGTGGGACAGAAGTT	47.2	615
P15	F: TGCCTGGTTCAAAGTCAC R: AGCTCTGGGAAATGGTGA	48.4	823
P16	F: ACTGTTGAGCCTGTTGGAGA R: AGGGGATTGAGTCAGGGATA	49.3	743
g.7254T>C	inner-F: ACCTAATGATTTCATGCACTGTTATGAC inner-R: ATGGATATAAGGATGCCTGTCAGCA outer-F: TGCCTAGTACAATTCTATCA outer-R: AACCCACTACAACTTCCTTCT	Touchdown-PCR	C allele: 364 T allele: 208 Outer primers: 521

Note: P means potential SNP site of goat *Boule* gene.

**Table 2 animals-09-00910-t002:** The tetra-primer amplification refractory mutation system PCR (T-ARMS-PCR) reaction system.

Composition		Volume (μL)					
Ratio of the inner and outer primers		1:1	1:2	**1:4**	1:6	1:8	1:10
Total volume		10	10	**10**	10	10	10
2 × Taq PCR MasterMix		5	5	**5**	5	5	5
DNA		0.5	0.5	**0.5**	0.5	0.5	0.5
ddH_2_O		3.7	3.3	**2.5**	2.1	0.9	0.1
Inner primers	forward	0.2	0.2	**0.2**	0.2	0.2	0.2
	reverse	0.2	0.2	**0.2**	0.2	0.2	0.2
Outer primers	forward	0.2	0.4	**0.8**	1.2	1.6	2
	reverse	0.2	0.4	**0.8**	1.2	1.6	2

**Table 3 animals-09-00910-t003:** Genotypic and allelic frequencies and population indexes of *Boule* gene in the Shaanbei White-Cashmere goat (SBWG).

Groups/Locus	Sizes	Genotypic Frequencies			Allelic Frequencies		^a^ HWE	Population Parameters			
SNP	N	CC	CT	TT	C	T	*p* Values	^b^ Ho	^c^ He	^d^ Ne	^e^ PIC
female goats with single-lamb	176	0.051 (*n* = 9)	0.392 *(n* = 69)	0.557 (*n* = 98)	0.247	0.753	*p* = 0.478	0.628	0.372	1.593	0.303
female goats with multi-lamb	181	0.061 (*n* = 11)	0.536 (*n* = 97)	0.403 (*n* = 73)	0.329	0.671	*p* = 0.004	0.559	0.441	1.790	0.344
Total (female goats with single-lamb + multi-lamb)	357	0.056 (*n* = 20)	0.465 (*n* = 166)	0.479 (*n* = 171)	0.288	0.712	*p* = 0.012	0.589	0.411	1.697	0.411

Note: SBWG, Shaanbei White-Cashmere goat. ^a^ HWE, Hardy–Weinberg equilibrium. ^b^ Ho, observed homozygosity. ^c^ He, heterozygosity. ^d^ Ne, effective allele numbers. ^e^ PIC, polymorphism information content.

**Table 4 animals-09-00910-t004:** Chi-square (χ^2^) and *p* values from genotype and allele frequencies among different groups at the single nucleotide polymorphism (SNP) locus of the *Boule* gene in the SBWG.

Column Header	Single-Lamb	Multi-Lamb
**Single-lamb**		**χ^2^ = 8.510** **(df = 2, * *p* = 0.014)**
**Multi-lamb**	χ^2^ = 2.892 (df = 1, *p* = 0.089)	

χ^2^ (*df, p value*) represents the differences of genotype frequencies between two groups in the up-triangle; χ^2^ (*df, p value*) represents the difference of allele frequencies between two groups in the down-triangle. Continuous correction was only performed for the 2 × 2 contingency table. * *p* < 0.05.

**Table 5 animals-09-00910-t005:** Relationship between the SNP of the *Boule* gene and the litter size of the SBWG.

Growth Traits	Genotype (Mean ± SE)			*p* Value
	CC (*n*)	CT (*n*)	TT (*n*)	
**Litter size**	^a^ 1.55 ± 0.114 (*n* = 20)	^a^ 1.60 ± 0.041 (*n* = 166)	^b^ 1.46 ± 0.042 (*n* = 171)	*p* < 0.05

SBWG, Shaanbei White-Cashmere goat; the same letter indicates no significant difference (*p* > 0.05) and the different letter indicates significant difference (*p* < 0.05).

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
