# Peer review of "Polymorphisms within the Boule Gene Detected by Tetra-Primer Amplification Refractory Mutation System PCR (T-ARMS-PCR) Are Significantly Associated with Goat Litter Size"

_animals, 2019, doi:10.3390/ani9110910_

Round 1
Reviewer 1 Report
Line 114-115: “All the individuals used were in the same age group” should be more detail. How old is the goats sampled in this study? Line 184: “breeds” or population or subpopulation? Only one breed SBWC was observed. Since there were some other potential markers for SBWC, as mentioned in the section of introduction, it is a big challenge that how to control their effects. Boule gene is a main cause of sperm deficiency, so it would be better to consider the effects of genotype in male animals. What’s more, more complex model containing some important factors such as male effects, farm, season are recommended if possible.
Author Response
Dear Editor and anonymous reviewers,
We greatly appreciate the anonymous reviewers for your careful review and constructive comments of our manuscript (ID: animals-594400). We have studied comments carefully and tried our best to revise this manuscript, and we hope that the revision can meet with your approval.
Here, we have listed the point-by-point responses to your detailed comments and suggestions (with blue and red). As follows:
Comments (with black) from Reviewer 1 and author responses (with bule and red):
Question 1: Line 114-115: “All the individuals used were in the same age group” should be more detail. How old is the goats sampled in this study?
Answer 1:
Thanks a lot for your reminding. The goats used in this study are all at 2-3 years old. In the revised manuscript, the corresponding sentence in “Materials and methods” had also been modified as follows:
All the used individuals were in the same age group (2-3 years old).
Question 2: Line 184: “breeds” or population or subpopulation? Only one breed SBWC was observed.
Answer 2:
As you mentioned, only one breed SBWC was used in this study. Herein, all of the individuals were divided into two groups, including female goats with single-lamb and female goats with multi-lamb. So, in the revised version, all of the “breeds” were replaced by “groups”.
Question 3: Since there were some other potential markers for SBWC, as mentioned in the section of introduction, it is a big challenge that how to control their effects.
Answer 3:
As we all known, litter size is a quantitative trait controlled by multiple genes (Ma et al., 2018). Therefore, even though several genetic markers associated with lambing in SBWC have been identified, we still hope to discover a new potential molecular marker associated with litter size. So, the Boule gene, a gene closely related to mammalian reproduction and whose polymorphisms had not been explored in goats, were detected in this study.
Addtionally, the reported genes, such as mentioned GHR (chromosome 20), CMTM2 (chr.18), MARCH1 (chr.6), KDM6A (chr.X), and CTNNB1 (chr.22), are located far from the Boule gene (chr.2). Therefore, there is almost no possibility that those reported polymorphisms have a linkage relationship with the g.7254T>C of Boule gene.
Reference:
Ma X, Li PH, Zhu MX, He LC, Sui SP, Gao S, Su GS, Ding NS, Huang Y, Lu ZQ, Huang XG, Huang RH. Genome-wide association analysis reveals genomic regions on Chromosome 13 affecting litter size and candidate genes for uterine horn length in Erhualian pigs. Animal. 2018. 12(12): 2453-2461.
Question 4: Boule gene is a main cause of sperm deficiency, so it would be better to consider the effects of genotype in male animals.
Answer 4:
Considering that the modern farming of goats tends to be intensive, female goats have become the mainstay of farming, while only a small number of male goats are centrally managed in breeding farms to provide semen. Therefore, it is difficult to obtain many male goat samples for experimental research. As you mentioned that Boulegene is largely known as a main cause of sperm deficiency, but very little data exists concerning its potential function in female reproductive traits. So, a total of 357 female goats were detected in this study. Herein. g.7254T>C had a significant effect on litter size in the SBWG breed, speculating that the Boule gene might underlie female reproductive capacities.Question 5: What’s more, more complex model containing some important factors such as male effects, farm, season are recommended if possible.
Answer 5:
In this study, the adjusted linear model with fixed effects was used to deal with the relationships between genotypes and litter size of 357 goats. The adjusted model included fixed effects of marker genotype, birth year, age of dam, sire, farm, sex, breed and random effects of animal.
Linear Model: Yijkl = μ + Bi + Aj + Gk + (BG)ik + Eijkl, where Yijkl was the trait measured on each of the ijklth animal; μ the overall mean; Bi the type of the ith breed; Aj the type of the jth age; Gk the type of the kth gentotype; (BG)ik, the interaction between the ith breed and the kth genotype, and Eijkl was the random error.
Herein, only one breed (SBWC) was analysed and all dams were at the same age (2-3 years old). So an effect associated with farm, sex and season of birth (spring versus fall), age of dam and sire were not into linear model, as the previous study indicated that these effect did not have significant influences on variability of traits in female populations (Lan et al., 2007). Therefore, the reduced model used in the final analysis is as follows: Yk = μ + Gk + Ek. Furthermore, as described in “Statistical analysis” part, c2 test and analysis of variance (ANOVA) available in statistical software SPSS (Version 18.0) were used to analyze the relationship between the genotypes and litter size in goat.
Reference:
Lan XY, Pan CY, Chen H, Zhang CL, Li JY, Zhao M, Lei CZ, Zhang AL, Zhang L. An AluI PCR-RFLP detecting a silent allele at the goat POU1F1 locus and its association with production traits. Small Ruminant Research. 2007. 73: 8-12.
We sincerely hope that the current revised version can be considered to be acceptable for publication in Animals.
With best regards!
Sincerely Yours,
Xiaoyue Song (songxiaoyue@yulinu.edu.cn),
Jie Li (lijie950302@163.com),
Xianyong Lan (lanxianyong79@126.com) (corresponding author),
Lei Qu (ylqulei@126.com) (corresponding author), et al.
College of Animal Science and Technology,
Northwest A&F University, Yangling, Shaanxi 712100, China

Reviewer 2 Report
The present work is in line with journal interest and in general is well written and presented. Regarding the general form of the manuscript my advice is a moderate English revision because some spelling and some colloquial sentenced are reported even if I am not in position to make advise about grammar. The work is interesting and focused on a poorly studied gene but in my opinion could be improved at phenotype recording: a more detailed and richer phenotype dataset it would have increased the quality of the paper. A part of this more attention should be taken to studied polymorphisms, for example about the first screening to discover them
The sentences: “72 Furthermore, except for indel mutation, the main type of genetic variation, the single nucleotide 73 polymorphism (SNP) is proposed to adversely affect mRNA stability or protein binding and74 eventually change the target traits [6,7].” Is referred to the genes in the previous paraph? Not all the SNPs affect RNA and not all the indel will not. Probably can be deleted de the information if superfluous here.
Line 78 “Another famous multiparous gene, goat GDF9,” I am doubtful about the correctness of the word multiparous, anyway it is not “pretty”
Line 79 “and the famous Q320P mutation was the major SNP” please avoid personalism …it is famous for you? and why? Change better for “highly used” or “highly cited” etc etc
Line 82-83:” except for the candidate gene,” what means? Canditate gene have not SNPs in the cited regins?
Line 85: change “economic trait” to “reproductive trait”
Line98: what does the author intended for “meiosis block”?
Line 123: the mutation g.7254T>C is firstly reported here. Please include in the methods some detail about the position or better the rs accession
Line 124 : the NC 030809.1 accession is referred to complete reference genome not to the gene. Pleaase report the correct accession number
Line 128: delete “Among various detection methods, the T-ARMS-PCR has been successfully used in many studies to detect SNPs and their biological markers [24]” or move to introduction or discussion.
Line 137-146: some papers are cited here as 4, 25, 26 and 27. What does the author want to cite? That the technique cited in the paper was used? In my opinion is not necessary in many cases, like the use of agarose gel to visualize genotypes.
Table 1: what do primers P1 to P16 stand for? They were used to sequence the gene? It is not clear in the text. If they are described polymorphism retrieved in literature they should be described before
Line 148: what does “potential” stand for?
line 152: in my opinion PIC can be omitted here. In SNPs it makes no sense
Table 3: what does the super index stand for in the abbreviation of the calculated parameters?
Line 203-205: table 5 is missing…. It seems that there are two tables 4. Please check
Line 220: I miss some information about repeatability of this technique in the work. The author report that sequencing was used. I would like more detail about the strongness of the technique
Line 222: the mutation was genotyped not identified, please correct
Line 223: what mutations?
Line 232: what does the author what to say with “introns has become more and more clear”? please clarifies
Line 237: please be more clear with definition…. meiosis gene is to much general. Use better gene that codifying or genes encoding etc
Author Response
Dear Editor and anonymous reviewers,
We greatly appreciate the anonymous reviewers for your careful review and constructive comments of our manuscript (ID: animals-594400). We have studied comments carefully and tried our best to revise this manuscript, and we hope that the revision can meet with your approval.
Here, we have listed the point-by-point responses to your detailed comments and suggestions (with blue and red). As follows:
Comments (with black) from Reviewer 2 and author responses (with bule and red):
Question 1: The present work is in line with journal interest and in general is well written and presented. Regarding the general form of the manuscript my advice is a moderate English revision because some spelling and some colloquial sentenced are reported even if I am not in position to make advise about grammar.
Answer 1:
Thanks a lot for your recognition of this study and for your suggestions. The language of this manuscript has been polished by MDPI, which was recommended by Animals. The grammar and logic have also been professionally modified by MDPI, but there may be some spelling errors. Therefore, in the revised version, the manuscript has been carefully checked and the corresponding words have been corrected.
Question 2: The work is interesting and focused on a poorly studied gene but in my opinion could be improved at phenotype recording: a more detailed and richer phenotype dataset it would have increased the quality of the paper. A part of this more attention should be taken to studied polymorphisms, for example about the first screening to discover them.
Answer 2:
Thank you. Considering the Boulegene closely related to animal reproduction, in this study, the used goat individuals were only recorded for age, health status and first birth litter size. According to your suggestion, the those information was also enriched in the revised version. The modified sentences are as follows:All the used individuals were in the same age group (2-3 years old) and healthy. Moreover, from the agricultural technical station of Hengshan county by production records, the data of the first birth litter size of Shaanbei Cashmere goats were also collected. According to the production records, the 357 goats were divided into a single-lamb group (n = 176) and a multi-lamb group (n = 181).
Additionally, the detected SNP mutation, rs661484476: g.7254T>C, was screened in the ensembl database, so the “first screening of this mutation site” is not rigorous. However, as described in the manuscript, it was the first time to experimentally verified the g.7254T>C and its association with reproduction was also explored.Question 3: The sentences: “72 Furthermore, except for indel mutation, the main type of genetic variation, the single nucleotide 73 polymorphism (SNP) is proposed to adversely affect mRNA stability or protein binding and74 eventually change the target traits [6,7].” Is referred to the genes in the previous paraph? Not all the SNPs affect RNA and not all the indel will not. Probably can be deleted de the information if superfluous here.
Answer 3: Thank you for your careful review. The previous sentence was unreasonable and rigorous. But as a transitional sentence, its existence is logically critical. Therefore, this sentence has been modified as follows:
Furthermore, except for indel mutation, the main type of genetic variation, some single nucleotide polymorphism (SNP) are proposed to can adversely affect mRNA stability or protein binding and eventually change the target traits [7,8].
Question 4: Line 78 “Another famous multiparous gene, goat GDF9,” I am doubtful about the correctness of the word multiparous, anyway it is not “pretty”.
Answer 4: The spelling of “multiparous” is correct, and it means producing more than one offspring at a time.
Question 5: Line 79 “and the famous Q320P mutation was the major SNP” please avoid personalism …it is famous for you? and why? Change better for “highly used” or “highly cited” etc etc
Answer 5: Thanks for your suggestion. “famous” was changed by “highly used” in revised version sentence.
Question 6: Line 82-83:” except for the candidate gene,” what means? Canditate gene have not SNPs in the cited regins?
Answer 6: Thanks for your remind. The sentence was changed as follow: Furthermore, SNPs existed around genome flanking regions surrounding the transcription start sites also can contributed to enhancing goat litter size [15].
Question 7: Line 85: change “economic trait” to “reproductive trait”.
Answer 7: Thanks for your advice. In this sentence, “economic trait” was changed to “reproductive trait”.
Question 8: Line98: what does the author intended for “meiosis block”?
Answer 8: The sentence “Only some SNPs in introns are reported in humans with meiosis block, but not in goat...” indicated that the identified SNPs of Boule gene is only reported in human and associated with meiosis block of man.
Question 9: Line 123: the mutation g.7254T>C is firstly reported here. Please include in the methods some detail about the position or better the rs accession.
Answer 9: This sentence was removed in revised version for the detected SNP mutation, g.7254T>C, was screened in the ensembl database, so the “first screening of this mutation site” is not rigorous. In the results part, it described as follow:
One SNP locus (NC-030809.1, rs661484476: g.7254T>C), located at the second intron of goat Boule gene, was firstly detected in goat groups.
Question 10: Line 124 : the NC 030809.1 accession is referred to complete reference genome not to the gene. Pleaase report the correct accession number.
Answer 10: Thanks for your suggestion. The correct number of GeneID (102173977) was added.
Question 11: Line 128: delete “Among various detection methods, the T-ARMS-PCR has been successfully used in many studies to detect SNPs and their biological markers [24]” or move to introduction or discussion.
Answer 11: Thanks a lot for your suggestion. This sentence was deleted in method part.
Question 12: Line 137-146: some papers are cited here as 4, 25, 26 and 27. What does the author want to cite? That the technique cited in the paper was used? In my opinion is not necessary in many cases, like the use of agarose gel to visualize genotypes.
Answer 12: Thank you. According to your suggestion, some reference, such as previous 4, 27, were removed. But several mentioned cited papers, including 4 and previous 26, are still necessary for explanation of related backeground or related method.
Question 13: Table 1: what do primers P1 to P16 stand for? They were used to sequence the gene? It is not clear in the text. If they are described polymorphism retrieved in literature they should be described before.
Answer 13: As description in “materials and methods” part, those primers were designed to detect SNPs in the coding regions and noncoding regions of the Boule gene (Table 1) to detect potential mutations, and only one mutation (g.7254T>C) located at the second intron, was found and identified.
Question 14: Line 148: what does “potential” stand for?
Answer 14: It represents the mutation sites of Boule gene which can be screened in the ensembl database, but they still need to be identified by experimental research.
Question 15: line 152: in my opinion PIC can be omitted here. In SNPs it makes no sense.
Answer 15: I am sorry that on the issue of PIC in SNPs, we do not agree with you. Polymorphic formation content (PIC) can be defined as the frequency of a polymorphic marker in a population. It is generally used to measure genetic heterozygosity and also can be used to indicate the degree of polymorphism at a locus in a population. So values of PIC are necessary for evaluating a polymorphic marker, including SNP.
Question 16: Table 3: what does the super index stand for in the abbreviation of the calculated parameters?
Answer 16: Thanks for your remind. The super index in the abbreviation of the calculated parameters are meaningless. So they were removed.
Question 17: Line 203-205: table 5 is missing…. It seems that there are two tables 4. Please check
Answer 17: Thanks a lot for your careful review. It was changed as follows:
Table 5. Relationship between the SNP of the Boule gene and the litter size of the SBWG.
Question 18: Line 220: I miss some information about repeatability of this technique in the work. The author report that sequencing was used. I would like more detail about the strongness of the technique.
Answer 18:
The main principle of the T-ARMS-PCR method is that Taq DNA polymerase lacks the 3′-5′exonuclease activity, and if the primer's 3′ end is mismatched hence, its amplification rate is slower or it fails (Zhang et al., 2015). This method has been successfully used in SNPs detection due to its inexpensive, easy-to-operate, rapid, and accurate characteristics. For instance, T-ARMS-PCR method revealed the incidence of foot and mouth diseases in Frieswal cattle population (Singh et al., 2014), and polymorphisms of TNF-α gene promoter was identified being associated with chronic pancreatitis by T-ARMS-PCR (Sri Manjari et al., 2014).
In this study, as mentioned in “Materials and methods”, to confirm the mutation sites, DNA sequencing was performed only when products with different genotypes were firstly amplified. Subsequent genotyping of individuals were based on the results of the T-ARMS-PCR.
Reference:
Zhang SH, Dang YH, Zhang Q, Qin Q, Lei CZ, Chen H, Lan XY. Tetra-primer amplification refractory mutation system PCR (T-ARMS-PCR) rapidly identified a critical missense mutation (P236T) of bovine ACADVL gene affecting growth traits. Gene. 2015; 559(2): 184-188. Singh R, Deb R, Singh U, Alex R, Kumar S, Chakraborti S, Sharma S, Sengar G, Singh R. Development of a tetra-primer ARMS PCR-based assay for detection of a novel single-nucleotide polymorphism in the 5' untranslated region of the bovine ITGB6 receptor gene associated with foot-and-mouth disease susceptibility in cattle. Arch Virol. 2014; 159(12): 3385-3389. Sri Manjari K, Jyothy A, Shravan Kumar P, Prabhakar B, Uma Devi M, Ramanna M, Nallari P, Venkateshwari A. A single-nucleotide polymorphism in tumor necrosis factor-α (-308 G/A) as a biomarker in chronic pancreatitis. Gene. 2014; 539(2):186-189.Question 19: Line 222: the mutation was genotyped not identified, please correct.
Answer 19: Thank you. It was changed as “genotyped” in manuscript.
Question 20: Line 223: what mutations?
Answer 20: As mentioned in “materials and methods” part, some potential mutations in the coding regions and noncoding regions of the Boule gene were screened from Ensembl database, and related primers were designed to detect them (shown in Table 1). The sentence in line 223 was also modified as follows:
Herein, mutations in the coding region of the Boule gene were not found, but one SNP locus (g.7254T>C) was detected at intron 2.
Question 21: Line 232: what does the author what to say with “introns has become more and more clear”? please clarifies
Answer 21: The previous expression is ambiguous, so we modified this sentence as follows:
In recent years, the function of introns has been continuously revealed, especially in the intron-mediated splicing process [34-36].
Question 22: Line 237: please be more clear with definition…. meiosis gene is to much general. Use better gene that codifying or genes encoding etc
Answer 22: Thank you. According to your suggestion, this setence was changed as follows:
The Boule gene was found to be expressed in a similar pattern with gene of DNA meiotic recombinase 1 and mutS homolog 4 between 49 and 94 days postcoitum in the sheep ovary, suggesting its possible participation in female reproductive capacities [40].
We sincerely hope that the current revised version can be considered to be acceptable for publication in Animals.
With best regards!
Sincerely Yours,
Xiaoyue Song (songxiaoyue@yulinu.edu.cn),
Jie Li (lijie950302@163.com),
Xianyong Lan (lanxianyong79@126.com) (corresponding author),
Lei Qu (ylqulei@126.com) (corresponding author), et al.
College of Animal Science and Technology,
Northwest A&F University, Yangling, Shaanxi 712100, China

Round 2
Reviewer 2 Report
Dear author,
thanks for correcting the article and taking my advice and corrections into consideration. The article has been greatly improved and is in my opinion in the state of being published. Despite continuing disagreement with the author over the PIC, it is only my scientific opinion and not relevant to the article. Indeed following the official definition of PIC is : "The polymorphism information content of a marker is the probability that the marker genotype of the offspring of a heterozygous parent affected with a dominant disease allows one to deduce which marker allele the offspring inherited from the parent. It is a measure of a marker's usefulness for linkage analysis." So in my opinion a marker with good PIC is a good marker for parentage analysis or linkage studies but not necessarily for association and gene assisted selection studies. Then there is the problem that SNPs due to their nature are always moderately informative markers (the maxium is 0,375 when MAF is 0.5)
This manuscript is a resubmission of an earlier submission. The following is a list of the peer review reports and author responses from that submission.